# Determinant of Osteoporosis Preventive Behaviors among Perimenopausal Women: A Cross-Sectional Study to Explore the Role of Knowledge and Health Beliefs

**DOI:** 10.3390/nu15133052

**Published:** 2023-07-06

**Authors:** Wafaa T. Elgzar, Mohammed H. Nahari, Samiha H. Sayed, Heba A. Ibrahim

**Affiliations:** 1Department of Maternity and Childhood Nursing, Nursing College, Najran University, Najran 66441, Saudi Arabia; 2Department of Clinical Laboratory Sciences, College of Applied Medical Sciences, Najran University, Najran 66441, Saudi Arabia; 3Department of Community Health Nursing, Faculty of Nursing, Damanhour University, Damanhour 22516, Egypt; 4Department of Public Health, College of Health Sciences, Saudi Electronic University, Riyadh 11673, Saudi Arabia

**Keywords:** osteoporosis, health behaviors, perimenopause, knowledge, health beliefs

## Abstract

Osteoporosis is a silent chronic disease, and many people did not discover it until they were diagnosed with a fracture. Therefore, regular scanning and appropriate Osteoporosis Preventive Behaviors (OPB) are the management cornerstone. OPB is strongly affected by personal knowledge and health beliefs. This study explores the role of knowledge and health beliefs as determinants of OPB among perimenopausal women. This cross-sectional study was performed on 1075 perimenopausal women in Najran City, Saudi Arabia, from January 2023 to March 2023. The data collection instrument is a self-reported questionnaire consisting of basic data, the OPB scale, an osteoporosis knowledge assessment tool, and the osteoporosis health belief scale. The current study results showed that approximately one-quarter (27.8%) of the study participants had high OPB with an overall mean of 20.83 ± 5.08 grade. The group practicing high OPB demonstrated a higher knowledge mean (11.37 ± 2.99) than the low OPB group (9.93 ± 3.51). In addition, all health beliefs constructs significantly differed among the low and high OPB groups (*p* ˂ 0.05). The participant’s age, occupational status, educational level, attendance of training courses, and history of bone fractures were significantly associated with high OPB. Osteoporosis knowledge, perceived susceptibility, perceived seriousness, exercises’ perceived benefits, and health motivations are positive predictors of high OPB (*p* ˂ 0.05). The study concluded that osteoporosis-related knowledge and health beliefs—especially perceived susceptibility, perceived seriousness, exercises’ perceived benefits, and health motivations—are positive predictors of high OPB. The health belief model can be an effective tool used to determine high-risk groups who practice low OPB and build need-based educational interventions.

## 1. Introduction

Osteoporosis is a main health problem characterized by a decreased bone density of less than 2.5 standard deviations compared to an average healthy population of the same age and sex, disruption of bone, and susceptibility to fracture [1]. Although osteoporosis is seen in all age groups and genders, it is more common in women and mostly affects postmenopausal women [2]. According to the World Health Organization (WHO), around 200 million women have osteoporosis worldwide, and the prevalence varies by country, from 4.1% in the Netherlands to 52% in Turkey [3,4]. Osteoporosis prevalence is much higher in the Middle East than in Western countries [5]. There is a great variety of osteoporosis prevalence in Saudi Arabia. Epidemiological studies showed that the osteoporosis rate ranged from 8% to 63.8% [6,7]. Osteoporosis is a silent chronic illness, and many people did not discover it until they were diagnosed with a fracture. Therefore, regular scanning and checkup after age 45 are essential for early detection. Common symptoms include back pain, loss of height over time, and bones that fracture much easier than normal [8].

Osteoporosis is a multifactorial disease that includes several modifiable and non-modifiable factors. Modifiable risk factors include lifestyle factors, such as low dietary calcium and vitamin D deficiency, decreased sun exposure, inadequate exercise, smoking, and alcohol intake. The non-modifiable risk factors include old age, gender, and heredity. Increasing bone density during adolescence and adulthood can significantly decrease bone loss in old age and prevent osteoporosis [8]. Osteoporosis results when the maximum bone density and mass cannot reach their peak during the bone development age; therefore, it is not a normal part of aging. It occurs when the bone cannot withstand the anticipated normal bone loss during advanced age. Though osteoporosis may affect both sexes, 80% of sufferers are females. The higher osteoporosis prevalence in females may be attributed to many factors. First, females have lighter, thinner, and lower bone density than males [9]. Second, pregnancy and lactation decreased bone deposition in bone [10]. Third, estrogen is an important hormone in bone metabolism and turnover regulation; therefore, estrogen deficiency is a very important risk factor for osteoporosis during the postmenopausal period [11].

A healthy and active lifestyle for young females is the cornerstone in preventing and reducing osteoporosis prevalence and burden [1,12]. Prevention begins in childhood and extends throughout life by building strong bones and following a healthy lifestyle such as increasing weight-bearing physical activity, maintaining an optimal Body Mass Index (BMI), sufficient intake of calcium and vitamin D, sun exposure, quitting smoking and alcohol, and moderate caffeine and salt intake [13,14]. Therefore, osteoporosis is preventable through predicting and managing its modifiable risk factors. Increasing public awareness about osteoporosis risk factors and preventive measures may effectively decrease its prevalence. In addition, population health beliefs toward osteoporosis can enhance more positive preventive behaviors [15,16].

To get the benefits of health belief change in OPB, it’s essential to understand a person’s health beliefs toward osteoporosis to identify modifiable risk factors and enhance a healthy lifestyle. One of the most important theoretical models which explain engaging in protective behaviors is the Health Belief Model (HBM); this model anticipates that beliefs about specific diseases and their protective health behaviors influence the likelihood of practicing these preventive behaviors. These HB involve the individuals’ belief that they are susceptible to such condition (perceived susceptibility); understanding the severe complications which result from the disease (perceived severity); belief in the benefits of the preventive behavior (perceived benefits) relative cost of the behavior (perceived barriers) also recognizes that these benefits outweigh the expected barriers associated with protective behavior; Finally, believe that they have the motivation to live a healthy lifestyle and the ability to follow these protective behaviors (health motivations) [17,18,19].

Osteoporosis is a preventable but rarely curable disease; therefore, enhancing preventive behavior is the cornerstone of its management. To enhance OPB, it is essential to explore its determinant and identify the role of knowledge and health beliefs in enhancing OPB. Therefore, the current study aims to explore the role of knowledge and health beliefs as determinants of OPB among perimenopausal women. The availability of such data may improve the quality of health awareness programs and direct the Ministry of Health’s efforts toward high-risk groups. In addition, the HBM scale may be used as a diagnostic tool to identify groups’ need for further interventions.

## 2. Materials and Methods

The current cross-sectional study was conducted in Najran City, Saudi Arabia, which is the largest city in the Najran region. Among the female population in Najran, 136,090 women between the ages of 20 and 60 make up 53.6% of all females in the region and 23.9% of the entire population, according to the 2016 Saudi Demography Survey [20].

The sample size needed for the study was determined using the Epi-info calculator. The population size was 136,090, and it was assumed that 50% of the population had high OPB. The absolute precision was 5%, and the design effect was 1%. The sample size was calculated to achieve a 99.9% confidence interval. The sample size is 1067 perimenopausal women; after adding 5% for anticipated inconsistent or missing data, the sample size was 1120.

The researchers used a convenience sampling method to enroll approximately 1120 women from Najran City. The inclusion criteria include women in the pre-menopausal period, aged 40–55, who can read and write and are willing to participate in the study. Before the data entry data check was performed, 45 sheets were excluded because of an age less than 40 or more than 55 years, and data analysis was performed on 1075 sheets.

### 2.1. Data Collection Instruments

The researchers created a structured self-reported questionnaire based on the related literature. It involves four main parts: the first basic data, the second OPB scale, the third Osteoporosis Knowledge Assessment Tool (OKAT), and the fourth osteoporosis health belief scale (OHBS).

Basic data comprised age, residence, marital status, education, occupational status, monthly income, personal and family history of osteoporosis, fractures, and attendance of osteoporosis educational interventions.

The researchers developed the OPB scale to evaluate the preventive practices performed by the women after reviewing related literature [13,21,22]. It is composed of seven statements rated on a 5-point Likert scale listed as always (5), usually (4), sometimes (3), rarely (2), and never (1). The overall OPB scale score ranged from 7–35, where higher scores indicated higher OPB. According to her overall score, the participants were considered to have low (7–21) or high (22–35) OPB. The Cronbach alpha coefficient was used to test the reliability of this scale, and the results indicated good reliability with a value of (r = 0.82).

The OKAT was developed by Winzenberg et al. to assess women’s knowledge regarding osteoporosis. It comprises 20 dichotomous questions (true and false) with four basic categories: osteoporosis symptoms, risk factors; preventive measures such as physical exercises and a healthy diet; and treatment possibilities. A right answer scored “1”, and a wrong answer scored”0”. The overall OKAT score ranged from 0 to 20, with a higher score indicating higher knowledge [23]. The internal consistency for the Arabic form of OKAT (r = 0.824) [24].

OHBS was created by Kim et al. to assess women’s beliefs regarding osteoporosis. It includes susceptibility to osteoporosis (6 Item), seriousness (6 Item), exercises and calcium intake benefits (12 Item), barriers to exercising and taking calcium (12 Item), and health motivation (6 Item). The OHBS was rated on a 5-point Likert scale ranging from strongly disagree (1) to strongly agree (5). The total score ranged from 42–210, with higher scores indicating higher health beliefs [25]. According to Sayed-Hassan and Bashour, the OHBS reliability Arabic version was high (r = 0.806) [24].

Data collection started from January 2023 to March 2023. A structured online questionnaire was used to collect data through Google Forms. The authors created a one-page recruitment sheet shared with individuals and groups through social media, e.g., WhatsApp, Telegram, Facebook, Instagram, Twitter, and email. The sheet provided information about the study’s purpose, procedures, voluntary participation, confidentiality, and instructions for completing the questionnaire. It also included a link to the online questionnaire. All the online questionnaire items are made as required to ensure data completeness.

### 2.2. Ethical Approval

The proposal and data collection instruments were assessed by the Najran Health Affairs ethical committee (IRB NO 2023-14 E) following clearance by the deanship of scientific research at Najran University. Before data collection, each participant provided their informed consent, and participant anonymity was applied. The electronic questionnaire began with a statement of informed consent, and it was necessary to accept it before continuing. Participants were informed of their freedom to decline participation without repercussions and that the collected information would only be utilized for research reasons.

### 2.3. Data Analysis

Data analysis was conducted via the Statistical IBM software, version 20 (IBM Corp., Armonk, NY, USA). The current study sample was conducted in a large sample; therefore, the Kolmogorov–Smirnov test was used to examine the normality of the data. The data was considered as normally distributed when *p* > 0.05. Descriptive statistics were performed to represent descriptive data. Among the study variables, residence, occupation, monthly income, history of osteoporosis, and bone fracture were categorical. Osteoporosis knowledge, preventive behaviors, and health beliefs were numerical variables. The total osteoporosis knowledge, preventive behaviors, and health beliefs were obtained by summing items. The differences between groups were checked using chi-square, Fisher’s Exact Test (FET), and the *t*-test. Binary logistic regression analysis was performed to examine the predictors of high OPB. The significant level was considered at *p* ˂ 0.05, and a 95% confidence level was adopted in all analyses. The Cox and Snell fitness test was used to check the total model goodness.

## 3. Results

### 3.1. Participants’ Basic

The mean age of the participants was 43.23 years, 46.1% had a normal weight (BMI = 18.5–25), and the majority (87.7%) were urban residents. Around two-thirds of the participants were employed (66.6%) and had a university education (69.9%). Less than half of them (45.3%) reported that their monthly income was sufficient, and they saved from it, and most of them (91.6%) were married. Approximately one-quarter (26.0%) received educational interventions about osteoporosis. Only 13.0% had personal history, 25.8% had a family history of osteoporosis, and 19.6% had a history of bone fracture. By comparing the basic data between participants with low and high OPB, the findings showed statistically significant differences (*p* ˂ 0.05) in age, occupational status, education, education, educational interventions, personal osteoporosis history, and bone fracture. However, there was no statistically significant difference (*p* > 0.05) regarding BMI, residence, monthly income, marital status, and family history of osteoporosis, as shown in Table 1.

### 3.2. Osteoporosis Preventive Behaviors

The mean scores and standard deviation of the OPB scale among perimenopausal women are illustrated in Table 2. The overall mean OPB score was 20.83 ± 5.08, with the mean score ranging from 2.51–4.05 for all OPB items. As revealed from the table, the highest mean was for non-smoking or alcoholic drinks (4.05 ± 0.95), and the lowest was for sunlight exposure (2.51 ± 1.07).

### 3.3. Perimenopausal Women’s Knowledge and Health Beliefs Are Differentiated by the Overall OPB Score

Table 3 illustrated statistically significant differences in perimenopausal women’s knowledge and all health beliefs constructs among the low and high OPB groups (*p* ˂ 0.05). The group practicing high OPB demonstrated a higher knowledge mean (11.37 ± 2.99) than the low OPB group (9.37 ± 3.51). Regarding health beliefs, the person practicing high OPB perceived themself as more susceptible (22.45 ± 6.53) to osteoporosis and perceived it as serious (23.59 ± 6.41) compared to the low OPB group (19.57 ± 4.59 and 20.03 ± 5.09), respectively. Moreover, women with high OPB had a higher perception of exercise (27.81 ± 3.12) and calcium intake (27.38 ± 3.11) benefits compared to the low group (22.68 ± 4.62) and (22.48 ± 4.30), respectively. On the contrary, women with high OPB perceive lower physical exercise (19.11 ± 5.22) and calcium intake (19.24 ± 5.01) barriers compared to the other group (22.86 ± 7.31 and 23.11 ± 7.34), respectively. Finally, women with high OPB had a higher health motivation (28.72 ± 1.59) than those with low OPB (21.63 ± 4.05).

### 3.4. Predictors of High OPB Using Logistic Regression Analysis

Table 4 explains the logistic regression analysis of predictors of high OPB. The participant’s age, occupational status, educational level, attendance of osteoporosis educational intervention, and history of bone fractures were significant predictors of high OPB. Older participants had a higher probability of practicing OPB 1.39 times for an increasing one year of age (AOR = 1.390; 95% CI 1.026–1.615, *p* = 0.011). Moreover, employed participants had higher odds of practicing OPB 1.5 times (AOR = 1.513; 95% CI 1.078–2.113, *p* 0.015) than housewives. A higher probability of practicing OPB was found in university-educated participants (AOR = 3.314; 95% CI 0.986–11.117, *p* 0.048) compared to the participants who read and wrote. Furthermore, a participant who did not receive osteoporosis educational intervention had decreased odds (AOR 0.651; 95% CI 0.476–0.877, *p* 0.003) for practicing high OPB than the participant who received an educational intervention. In addition, the participants who did not have a history of a bone fracture had a lower probability of practicing OPB (AOR = 0.715; 95% CI, 0.529–0.989, *p* 0.046) when compared with the participant with a positive history. The table also shows osteoporosis knowledge, perceived susceptibility, perceived seriousness, exercises’ perceived benefits, and health motivations are positive predictors of high OPB (*p* ˂ 0.05). An increase in one grade in the osteoporosis-related knowledge increased women’s probability to practice high OPB by 1.6 times (AOR = 1.607; 95% CI = 1.012–1.620, *p* = 0.035). Regarding health beliefs, one grade increased in perceived susceptibility (AOR = 1.671; 95% CI = 1.037–1.707, *p* = 0.001) and perceived seriousness (AOR = 1.345; 95% CI = 1.057–1.320, *p* = 0.000) increased women’s probability to practice high OPB by 1.6 and 1.3, respectively. In addition, one grade increased in perceiving exercise benefits (AOR = 1.371; 95% CI = 1.342–1.446, *p* = 0.000) and health motivation (AOR = 2.522; 95% CI = 2.182–2.869, *p* = 0.000) increased women’s probability to practice high OPB by 1.3 and 2.5 times, respectively. The Nagelkerke R Square test illustrated that 63% of the probability of practicing high OPB could be predicted through the current model.

## 4. Discussion

Many factors can affect bone strength throughout a person’s life. Some of these factors are not modifiable such as age, gender, family history, and race, and some are modifiable such as diet, physical activity, and sun exposure [26]. Therefore, high-risk people must have adequate knowledge and sound beliefs about osteoporosis to take preventive measures against this disease [27]. The current study reported low overall OPB among over two-thirds of perimenopausal participants. This result was not surprising because good and sufficient behavior is typically based on sufficient knowledge and beliefs, which were generally deficient among low OPB participants. Although various studies have documented that 15 min of direct sun exposure each day is a critical step in preventing osteoporosis, this item was the lowest on average among the current study participants. Lack of exposure to sunlight may be attributed to the hot climate in most regions of Saudi Arabia. At the same time, a sedentary lifestyle and socioeconomic well-being also contributed to a lack of sun exposure and regular physical activity or walking at least half an hour per day. These findings are in line with a prior Saudi study in which most female participants reported no sun exposure and low physical exercise [28]. Another Saudi study compared male and female adherence to a healthy lifestyle; they reported insufficient adherence to a healthy lifestyle and OPB among their female participants [29]. On the other side, our study’s results contradict the Egyptian study’s results [27], revealing that Egyptian females are usually exposed to sunlight daily. This difference may be attributed to social and cultural factors. In the Egyptian study, the majority of the participants were housewives who regularly engaged in outdoor activities, which helped them get enough vitamin D from exposure to sunlight. In the current study, most participants were employees with high socioeconomic status and a sedentary lifestyle. In addition, in Egypt, the climate temperature is tolerable in most Egyptian regions, and most females walk daily in the sun for work or home-related activities.

Concerning the predictors of high OPB, the current findings indicated that older and working participants with higher education, who attended osteoporosis educational intervention and had a history of bone fractures, had a higher probability of practicing high OPB. On the same line, Tan et al. evaluated osteoporosis awareness in Singapore. They reported that patients with low educational levels or no formal education had a significantly higher probability of low osteoporosis awareness and OPB [30]. Our findings were consistent with another study that documented a significant association between age and educational level with engagement in OPB [31]. The association between age and OPB may be attributed to the fact that many people believe that osteoporosis affects only older adults. As a result, young women may not be interested in learning more about the disease or adhering to its preventive behavior. It is important to consider these predictors when developing educational interventions, as osteoporosis prevention should begin in adolescence to ensure optimal bone density. Undoubtedly, educational interventions have a positive impact on the OPB. Several studies reported significant improvement in participants’ OPB after the implementation of educational interventions using HBM [32,33], the theory of planned behavior [34], and the protection motivation theory [35].

According to the current study findings, osteoporosis-related knowledge had a positive association with OPB. An increase in one grade in osteoporosis-related knowledge increased women’s probability of practicing high OPB by 1.6 times. In the same line as the present study, Kang and Kim tried to explore the predictors of OPB among women in their 20 s and 30 s. They elaborated that women with higher osteoporosis-related knowledge had significantly higher healthy behaviors and had at least one weight control experience [36]. Moreover, Rastgoo et al. conducted a study to investigate the predictors of OPB among adolescents and found an indirect effect of knowledge on OPB via the intermediary roles of attitudes and behavioral intention [37].

On the contrary, Wallace documented that although 65% of their participants could correctly answer questions related to osteoporosis knowledge, only 30% of them stuck to the recommended activity level and calcium intake. They added that knowledge did not significantly predict OPB [38]. The differences between the current study and Wallace’s may be due to differences in the participants’ age; he studied university students in their 20 s, while the current study involved perimenopausal women. Young women may perceive themselves away from osteoporosis as it is linked to old age; therefore, they may ignore any knowledge related to osteoporosis prevention. Conversely, perimenopausal women perceive themselves as susceptible to menopause-related complications, including osteoporosis; therefore, they may use knowledge more effectively to improve their OPB. Knowledge is an essential element linked to health behavior. Yet, it is not enough by itself to enable or promote behavioral change as strong health beliefs, including perceived susceptibility [38].

Our study findings showed that the constructs of HBM and osteoporosis-related knowledge predict 38.6% of the variance in OPB. In other words, the HBM constructs significantly enhanced OPB among perimenopausal women. Therefore, developing HBM-based educational interventions that especially address perceived susceptibility, seriousness, benefits, and health motivation would improve OPB. The HBM is known widely as a motivating tool for healthy behaviors; therefore, it was used extensively as a framework for numerous interventional research [39,40,41]. For example, Evenson and Sanders investigated the impact of HBM-based education in improving Calcium and vitamin D intake. They reported that the total health belief score significantly predicted starting and maintaining calcium and vitamin D intake [42]. In addition, Wallace reported that HBM is a strong tool that can predict OPB. They further elaborated that osteoporosis, perceived susceptibility, and health motivation were significant positive predictors for OPB, and barriers to exercise and calcium intake were negative predictors [38]. The relationship between health beliefs and preventive behaviors is interesting and old; however, still controversial. McLeod and Johnson conducted a systemic review to evaluate 22 studies to draw a conclusion about osteoporosis health beliefs using the health belief osteoporosis scale and self-efficacy scale. They found that perceived seriousness, calcium and exercise perceived benefits, self-efficacy and health motivation, and lower perceived barriers were the most important predictors for OPB. In addition, the total health belief scores were significantly associated with high OPB; therefore, they further recommended that using HBM as a framework for osteoporosis public awareness programs may be an effective intervention to change and promote OPB [43]. In addition, Self-efficacy and health motivation were found to be the most important motivators for OPB, mainly physical activities, in many studies [36,44]. Moreover, an Iranian study found a significant association between walking behaviors and the perceived susceptibility to osteoporosis and health motivation. In addition, healthy nutritional practices were positively associated with perceived susceptibility and self-efficacy while negatively associated with perceived barriers. The study added that HBM could predict 29.1% of the walking behaviors and 20.2% of the nutritional behaviors, and in conclusion, HBM could predict OPB [31]. Other experimental studies applied health belief model-based interventions to initiate and promote OPB and followed their participants for one year. They reported that HBM-based intervention induced change in the participants’ behaviors even one-year post-intervention [33].

## 5. Study Strengths and Limitations

There are several strengths to the current research. Including a sizable number of participants ensures that we have enough statistical power to examine the impact of different factors on OPB. In addition, this is the first Saudi study to assess the role of knowledge and health beliefs on OPB using HBM. The current study used standardized instruments for data collection such as OKAT and HBMS. However, using self-reported questionnaires to evaluate knowledge and beliefs may result in participants exaggerating or minimizing their responses. Additionally, the current study used a convenience sampling technique to recruit participants from social media, which may limit the generalizability of the findings.

## 6. Conclusions

The findings revealed low overall OPB among more than two-thirds of perimenopausal participants. Therefore, the health authority should be concerned about these results as they increase the risk of osteoporosis in women. High OPB was mostly noted among older and working participants with higher education who attended osteoporosis educational interventions and had a previous history of bone fractures. Moreover, osteoporosis-related knowledge, perceived susceptibility, perceived seriousness, exercises’ perceived benefits, and health motivations are positive predictors of high OPB. Therefore, developing HBM-based educational interventions that especially address perceived susceptibility, seriousness, benefits, and health motivation would improve OPB. Further study that compares bone mineral density data between high and low OPB is recommended.

## Figures and Tables

**Table 1 nutrients-15-03052-t001:** Perimenopausal women’s basic data were differentiated by the overall OPB score (*n* = 1075).

Variables	Cohort *n* = 1075		OPB				**χ^2^/FET/t**	***p*-Value**
			Low *n* = 773		High *n* = 302			
	n	%	*n*	%	*n*	%		
**Age (years) mean ± SD**	**43.23 ± 5.75**		42.88 ± 5.50		44.12 ± 6.25		3.182	0.002 *
**BMI**							6.356	0.096
- Underweight (BMI < 18.5)	53	4.9	41	5.3	12	4.0		
- Normal weight (BMI = 18.5 < 25)	492	45.8	369	47.7	123	40.7		
- Overweight (BMI = 25 < 30)	333	31.0	232	30.0	101	33.4		
- Obese (BMI = 30 and more)	190	17.7	127	16.4	63	20.9		
- Missing values	7	0.7	4	0.5	3	1.0		
**Residence**							0.364	0.546
- Rural	132	12.3	92	11.9	40	13.2		
- Urban	943	87.7	681	88.1	262	86.8		
**Occupational status**							4.586	0.033 *
- Housewife	359	33.4	273	35.3	86	28.5		
- Employee	716	66.6	500	64.7	216	71.5		
**Education**							11.875	0.003 *
- Read and write	19	1.8	14	1.8	5	1.7		
- Secondary education	305	28.4	242	31.3	63	20.9		
- University education	751	69.9	517	66.9	234	77.5		
**Monthly income**								
- Sufficient and save	487	45.3	351	45.4	136	45.0	3.743	0.154
- Sufficient	317	29.5	238	30.8	79	26.2		
- Insufficient	271	25.2	184	23.8	87	28.8		
**Marital status**							4.716	0.194
- Married	985	91.6	707	91.5	278	92.1		
- Single	38	3.5	24	3.1	14	4.6		
- Divorced	25	2.3	22	2.8	3	1.0		
- Widowed	27	2.5	20	2.6	7	2.3		
**Attendance of osteoporosis educational intervention**							8.308	0.004 *
- Yes	279	26.0	182	23.5	97	32.1		
- No	796	74.0	591	76.5	205	67.9		
**Personal history of osteoporosis**							5.212	0.022 *
- Yes	140	13.0	112	14.5	28	9.3		
- No	935	87.0	661	85.5	274	90.7		
**Family history of osteoporosis**							3.010	0.083
- Yes	277	25.8	188	24.3	89	29.5		
- No	798	74.2	585	75.7	213	70.5		
**History of a bone fracture**								
- Yes	211	19.6	164	21.2	47	15.6	4.399	0.036 *
- No	864	80.4	609	78.8	255	84.4		

χ^2^: Chi-square; FET: Fisher’s Exact Tests; * significant at *p* ˂ 0.05.

**Table 2 nutrients-15-03052-t002:** Mean scores and standard deviation of OPB scale among perimenopausal women.

OPB Scale	Mean	SD
Sunlight exposure 15 min/day	2.51	1.07
Consumption of food-containing calcium and vitamin D.	2.89	0.98
Taking calcium and vitamin D supplementation.	2.96	0.95
No smoking or alcoholic drinks	4.05	0.95
Walking half an hour daily	3.04	0.92
Practicing regular physical exercise	2.77	1.04
Have a regular health checkup	2.99	0.95
**The overall mean of the OPB score**	20.83	5.08

**Table 3 nutrients-15-03052-t003:** Perimenopausal women’s knowledge and health beliefs are differentiated by the overall OPB score.

Variables	Maximum Score	Cohort		OPB				*t*	*p*-Value
				Low		High			
		Mean	SD	Mean	SD	Mean	SD		
**Osteoporosis knowledge**	20	9.93	3.49	9.37	3.51	11.37	2.99	8.715	0.000 **
**Osteoporosis health beliefs**									
- Perceived susceptibility	30	20.38	5.36	19.57	4.59	22.45	6.53	8.117	0.000 **
- Perceived seriousness	30	21.03	5.72	20.03	5.09	23.59	6.41	9.552	0.000 **
- Exercise’s perceived benefits	30	24.12	4.84	22.68	4.62	27.81	3.12	17.748	0.000 **
- Calcium intake perceived benefits	30	23.85	4.57	22.48	4.30	27.38	3.11	18.022	0.000 **
- Physical exercise perceived barriers	30	20.16	6.122	22.86	7.31	19.11	5.22	9.388	0.000 **
- Taking calcium perceived barriers	30	20.33	6.01	23.11	7.34	19.24	5.01	9.877	0.000 **
- Health Motivation	30	23.62	4.76	21.63	4.05	28.72	1.59	29.528	0.000 **

*t*: Independent *t*-tests; ** significant at *p* ˂ 0.001.

**Table 4 nutrients-15-03052-t004:** Predictors of high OPB using logistic regression analysis.

Predictors	High OPB	
	AOR (95% CI)	*p*-Value
**Age** (years)	1.390 (1.026–1.615)	0.011 *
**BMI**		0.421
- Underweight (BMI < 18.5)	Ref	
- Normal weight (BMI = 18.5 < 25)	0.457 (0.083–2.528)	0.370
- Overweight (BMI = 25 < 30)	0.611 (0.271–1.380)	0.236
- Obese (BMI = 30 and more)	0.506 (0.242–1.145)	0.102
**Residence**		
- Rural	Ref	
- Urban	1.038 (0.398–2.710)	0.939
**Occupation**		
- Housewife	Ref	
- Employee	1.513 (1.078–2.113)	0.015 *
**Educational level**		0.017 *
- Read and write	Ref	
- Secondary education	1.239 (0.466–3.348)	0.662
- University education	3.314 (0.986–11.117)	0.048 *
**Monthly income**		0.308
- Sufficient and save	Ref	
- Sufficient	1.124 (0.521–2.425)	0.766
- Insufficient	0.654 (0.301–1.419)	0.283
**Marital status**		0.225
- Married	Ref	
- Single	1.134 (0.468–2.676)	0.774
- Divorced	1.670 (0.563–4.937)	0.349
- Widowed	0.388 (0.087–1.720)	0.224
**Attendance of osteoporosis educational intervention**		
- Yes	Ref	
- No	0.651 (0.476–0.877)	0.003 *
**Personal history of osteoporosis**		
- Yes	Ref	
- No	1.554 (0.983–2.512)	0.052
**Family history of osteoporosis**		
- Yes	Ref	
- No	1.185 (0.799–1.738)	0.423
**History of a bone fracture**		
- Yes	Ref	
- No	0.715 (0.529–0.989)	0.046 *
**Osteoporosis knowledge**	1.607 (1.012–1.620)	0.035 *
**Osteoporosis health beliefs**		
- Perceived susceptibilityW	1.671 (1.037–1.707)	0.001 *
- Perceived seriousness	1.345 (1.057–1.320)	0.000 **
- Exercise’s perceived benefits	1.371 (1.342–1.446)	0.000 **
- Calcium intake perceived benefits	1.001 (0.919–1.087)	0.978
- Physical exercise perceived barriers	0.939 (0.865–1.008)	0.074
- Taking calcium perceived barriers	0.986 (0.974–1.052)	0.872
- Health Motivation	2.522 (2.182–2.869)	0.000 **
- 2 Log likelihood (337.211)	Cox & Snell R Square (0.581)	Nagelkerke R Square (0.637)

AOR: Adjusted Odd Ratio; CI: Confidence Interval; * significant at *p* ˂ 0.05; ** significant at *p* ˂ 0.001.

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
