# Peer review of "Determinant of Osteoporosis Preventive Behaviors among Perimenopausal Women: A Cross-Sectional Study to Explore the Role of Knowledge and Health Beliefs"

_nutrients, 2023, doi:10.3390/nu15133052_

Round 1

Reviewer 1 Report

The manuscript by Wafaa T. Elgzar et al. claims that osteoporosis-related knowledge and health beliefs especially perceived susceptibility, perceived seriousness, exercises' perceived benefits, and health motivations, are positive predictors of high Osteoporosis Preventive Behaviors (OPB). The health belief model can be an effective tool used to determine high-risk groups who practice low OPB and build need-based educational interventions. The conclusion is interesting and exciting. However, the manuscript could be improved after minor revision if the following comments and questions are addressed.

1.      How many volunteers in the project? 1120? 1075? 769+299?

2.      The sample form of structured self-reported questionnaire including the first basic data, the second OPB scale, the third Osteoporosis Knowledge Assessment Tool (OKAT), and the fourth osteoporosis health belief scale (OHBS) could be added in supplemental data.

3.      If the BMD data (T-Score) of several volunteers in every group could be showed in the project, that is better for the conclusion.

The manuscript is understandable, however it is not fluent and the language should be improved.

Author Response

Replay for reviewer comments (1)

Dear reviewer, thanks for your valuable comments, which greatly improved the quality of the manuscript.  Note all reviewer's one comments highlighted in blue.

Comment

Authors response

Page

Paragraph number

1.      How many volunteers in the project? 1120? 1075? 769+299?

The calculated sample size is 1067 perimenopausal women; after adding 5% for anticipated inconsistent data, the sample size was 1120. Before the data entry data check was done, 45 sheets were excluded because of age less than 35 or more than 55 years, and data analysis was performed on 1075 sheets.

There are 7 missing responses that were not included in the first variable (BMI). We added the signal of missing values under table 1 based on your observation. Thank you very much.  The Cohort

n= 1075

Low OPB n=(773)

High OPB n= (302)

4

Page (3) and

Table (1) in page (4)

2.      The sample form of structured self-reported questionnaire including the first basic data, the second OPB scale, the third Osteoporosis Knowledge Assessment Tool (OKAT), and the fourth osteoporosis health belief scale (OHBS) could be added in supplemental data.

Was added in the supplementary materials

-

-

3.      If the BMD data (T-Score) of several volunteers in every group could be showed in the project, that is better for the conclusion.

It good Suggestion, but difficult for us to reach our participants again because of anonymous data. However, your valuable suggestion was added in the recommendation for further study.

10

Under the conclusion section

The manuscript is understandable, however it is not fluent and the language should be improved.

Done using grammarly premium software. The report will attached

Reviewer 2 Report

The HBM is well established to evaluate behaviors especially related to risk reduction of conditions. The major criticism of this paper is the eligibility criteria for the sample (age 35-55 years). There is no definition provided for perimenopausal and the age range starting with age 35 years is not consistent with perimenopause. While developing peak bone mass in younger years is protective for women, it is the transition to menopause with changes in estrogen that has the greatest impact on bone density. Generally  bone mineral density is not assessed until women reach the menopausal age (In US for White women 51 years) and data on onset of menopause in Saudia Arabia would be helpful to know, if different than US. Similarly, unless there are specific risk factors, education about health behaviors to preserve BMD most commonly occurs during the perimenopause and transition to menopause ages. The mean age of participants in this study was 43 years and without any menstrual history, it is very possible that many of the women may not have been perimenopausal. This criticism raises the issue of the relevance of the findings.

Author Response

Reviewer comment (2)

Dear reviewer, thanks for your valuable comments, which greatly improved the quality of the manuscript.  Note all reviewer's two comments highlighted in green.

Comment

Authors response

Page

Paragraph number

The HBM is well established to evaluate behaviors especially related to risk reduction of conditions. The major criticism of this paper is the eligibility criteria for the sample (age 35-55 years). There is no definition provided for perimenopausal and the age range starting with age 35 years is not consistent with perimenopause. While developing peak bone mass in younger years is protective for women, it is the transition to menopause with changes in estrogen that has the greatest impact on bone density. Generally  bone mineral density is not assessed until women reach the menopausal age (In US for White women 51 years) and data on onset of menopause in Saudia Arabia would be helpful to know, if different than US. Similarly, unless there are specific risk factors, education about health behaviors to preserve BMD most commonly occurs during the perimenopause and transition to menopause ages. The mean age of participants in this study was 43 years and without any menstrual history, it is very possible that many of the women may not have been perimenopausal. This criticism raises the issue of the relevance of the findings.

Dear reviewer, thanks for your valuable comment, which draws our attention to a serious typo error in the inclusion criteria (40-55 years of age). Based on a previous study, the mean age of menopause in Saudi Arabia is 48 years (AlDughaither et al., 2015). The current study included perimenopausal women, and perimenopause refers to the time from when changes in the menstrual cycle (missed periods or changes in amount of bleeding) or menopausal symptoms start to one year after the last menstrual period and usually starts in a woman's 40's  (Australian menopause association). We were concerned with collecting data from early perimenopausal women as the first step to explore osteoporosis preventive behaviors before menopause; therefore, we tried to include early perimenopausal women (mean 43 years). In the same line with your comment, "Unless there are specific risk factors, education about health behaviors to preserve BMD most commonly occurs during the perimenopause and transition to menopause ages." The current study aimed to explore the osteoporosis preventive behaviors in this high-risk group to provide a strong scientific base for future educational intervention to this vulnerable group.

3

2nd paragraph

Reviewer 3 Report

This paper explores the role of knowledge and health beliefs as de- 14 terminants of OPB among perimenopausal women.

It is a well written paper with an interesting topic focused on middle east environment.

I have several comments before the acceptance of this paper 

1) the paper should follow the STROBE guidelines. Please include the checklist into the revised version providing eveidene that the authors covered all the sections

2) the title should report the statmeent "a cross sectional study

3) the table should state "p value" instead only p

4) line 21 ..unit of measure must be reported. Please check all numbers into the manuscript.. they must report the unit of measure

5) please create a logistic regression model including in a single model alla Predictors with the target outcome the High OPBù

6) data analysis are not professional. the section of data analysis you have to describe accurately the normality test..which software.. I assume SPSS? Please state... and every single test that you applyied and the context.

Author Response

Reviewer comment (3)

Dear reviewer, thanks for your valuable comments, which greatly improved the quality of the manuscript. Note all reviewer's three comments highlighted in yellow.

Comment

Authors response

Page

Paragraph number

1) the paper should follow the STROBE guidelines. Please include the checklist into the revised version providing eveidene that the authors covered all the sections

Done

-

Attached with the replay for reviewer comments

2) the title should report the statmeent "a cross sectional study

A cross sectional study was added in the title

1

Title

3) the table should state "p value" instead only p

Done

All-over the results section

4) line 21 .unit of measure must be reported. Please check all numbers into the manuscript. they must report the unit of measure

Done

1

In abstract

5) please create a logistic regression model including in a single model alla Predictors with the target outcome the High OPBù

Done

6-7

Table 1

6) data analysis are not professional. the section of data analysis you have to describe accurately the normality test..which software.. I assume SPSS? Please state... and every single test that you applyied and the context.

Done

4

2nd paragraph

Reviewer 4 Report

The hypothesis of this study  was, that body composition phenotyping by MRI may be feasible and allows for risk stratification in OSO syndrome. The authors conclude that MRI-based biomarkers of bone, muscle and fat may be feasible for body composition phenotyping may therefore allow for targeted risk stratification and further cardio-metabolic risk assessment in suspected OSO syndrome. 

The introduction is well written , with adequate bibliographic references . however it is excessively long. The objective of the study is not clearly established.

The methodology is complete, widely described, which would allow the study to be carried out by another research group.

The results are clear expressed in tables and easy to understand. The graphical representation of any of these tables could help to understand the results. The discussion is adapted to the results obtained. The authors express the limitations and strengths of the study. One of the unexplained limitations is the joint validation of the instruments used.

Author Response

Reviewer comment (4)

Dear reviewer, thanks for your time and effort spent in our manuscript revision. However, we have some doubt regarding the research hypotheses wrote in the revision form “The hypothesis of this study was, that body composition phenotyping by MRI may be feasible and allows for risk stratification in OSO syndrome.“ As it is totally different from our study aim. Any way we appropriately responded to your valuable comments. Note all reviewer's four comments highlighted in Gray

Comment

Authors response

Page

Paragraph number

The introduction is well written, with adequate bibliographic references. However it is excessively long. The objective of the study is not clearly established.

Done

2

Introduction section

The methodology is complete, widely described, which would allow the study to be carried out by another research group.

We appreciate your valuable comments.

-

-

The results are clear expressed in tables and easy to understand. The graphical representation of any of these tables could help to understand the results. The discussion is adapted to the results obtained. The authors express the limitations and strengths of the study. One of the unexplained limitations is the joint validation of the instruments used.

WE PRESENT THE TOOLS RELIABILITY AS A FOLWING

-        The Cronbach alpha coefficient was used to test the reliability of this OPB scale, and the results indicated good reliability with a value of (r= 0.82).

-        The internal consistency for the Arabic form of OKAT (r = 0.824).

-        According to Sayed-Hassan & Bashour, the OHBS reliability Arabic version was high (r = 0.806)

3

Under the tools of data collection section
